# predPhogly-Site: Predicting phosphoglycerylation sites by incorporating probabilistic sequence-coupling information into PseAAC and addressing data imbalance

Sabit Ahmed[1☯]*, Afrida Rahman[1☯], Md. Al Mehedi Hasan[1], Md Khaled Ben Islam[2], Julia Rahman[1¤], Shamim Ahmad[3]

**1** Computer Science and Engineering, Rajshahi University of Engineering and Technology, Rajshahi, Bangladesh, **2** Computer Science and Engineering, Pabna University of Science and Technology, Pabna, Bangladesh, **3** Computer Science and Engineering, University of Rajshahi, Rajshahi, Bangladesh

☯ These authors contributed equally to this work.
¤ Current address: Institute for Integrated and Intelligent Systems, Griffith University, Brisbane, Australia
* sabit.a.sirat@gmail.com

**Data Availability Statement:** All relevant data are within the paper and its Supporting information files.

## Abstract

Post-translational modification (PTM) involves covalent modification after the biosynthesis process and plays an essential role in the study of cell biology. Lysine phosphoglycerylation, a newly discovered reversible type of PTM that affects glycolytic enzyme activities, and is responsible for a wide variety of diseases, such as heart failure, arthritis, and degeneration of the nervous system. Our goal is to computationally characterize potential phosphoglycerylation sites to understand the functionality and causality more accurately. In this study, a novel computational tool, referred to as predPhogly-Site, has been developed to predict phosphoglycerylation sites in the protein. It has effectively utilized the probabilistic sequence-coupling information among the nearby amino acid residues of phosphoglycerylation sites along with a variable cost adjustment for the skewed training dataset to enhance the prediction characteristics. It has achieved around 99% accuracy with more than 0.96 MCC and 0.97 AUC in both 10-fold cross-validation and independent test. Even, the standard deviation in 10-fold cross-validation is almost negligible. This performance indicates that predPhogly-Site remarkably outperformed the existing prediction tools and can be used as a promising predictor, preferably with its web interface at http://103.99.176.239/predPhogly-Site.

## Introduction

Post-translational modifications (PTM) refer to specific events after the translation stage, where the covalent inclusion of specific functional groups occurs in a protein [1]. These modifications have enormous impacts on biological processes and proteomic analysis, such as cellular signal transduction, subcellular localization, protein folding, protein degradation, and are

**Funding:** The author(s) received no specific funding for this work.

**Competing interests:** The authors have declared that no competing interests exist.

also responsible for various kinds of diseases [2]. Therefore, accurate identification and effective comprehension of PTM sites are significant for basic research in disease detection, prevention, and various drug developments [3]. Among the 20 standard constituent amino acid residues of cellular proteins, modifications at lysine residue (K) are commonly known as lysine PTM or K-PTM. According to the literature, several K-PTMs such as acetylation, crotonylation, ubiquitination, phosphoglycerylation, glycation, methylation, butyrylation, succinylation, biotinylation can be aided by these covalent modifications [4–8].

Lysine phosphoglycerylation is one of the reversible post-translational modifications, newly discovered in mouse liver and human cells [8, 9]. The formation of 3-phosphoglyceryl-lysine (pgK) takes place when primary glycolytic intermediate (1,3-BPG) interacts with particular lysine residues [8, 10]. A wide variety of diseases, including heart failure, arthritis, and various types of neurodegenerative disorders can be caused by this phosphoglycerylation. Metabolic labeling with substantial glucose indicates that it can be derived from glucose metabolism [9]. It has significant effects on glycolytic enzyme activities and can build up on cells with high glucose exposure [11]. Potential feedback mechanism that contributes to the creation and redirection of glycolytic intermediates to specific biosynthetic pathways is also established [8–11]. Concerning the crucial role of phosphoglycerylation in such biological processes, the effective way to characterize its functional aspects is to identify phosphoglycerylation sites with higher efficacy. Although high throughput experimental procedures to characterize phosphoglycerylation sites are known to achieve higher accuracy, computational methods are getting popularity as an effective alternative because of their laborsaving, time and cost-efficient characteristics.

Recent studies on identifying phosphoglycerylation sites have introduced several computational tools such as, Phogly-PseAAC [9], CKSAAP_PhoglySite [8], iPGK-PseAAC [12] and Bigram-PGK [11]. The first one has applied a KNN-based predictor with the pseudo amino acid feature source [9], where the second one has implemented a fuzzy SVM based predictor with the formation of k-spaced amino acid pairs feature set [8]. iPGK-PseAAC has utilized the pairwise coupling technique with an SVM classifier [11, 12]. The most recently developed predictor, Bigram-PGK has employed SVM with evolutionary information of the sequences for performance improvement [11]. Among these four predictors, only Bigram-PGK can predict phosphoglycerylation sites with an AUC higher than 0.90. However, the overall performance of this predictor needs further improvement in terms of other measurement metrics to be used as a complementary phosphoglycerylation site identification technique.

For constructing an efficient predictor, appropriate informative patterns connected with phosphoglycerylation need to be extracted. In this study, we are introducing a novel computational tool predPhogly-Site for predicting phosphoglycerylation sites by blending vectorized sequence coupling information with PseAAC [3, 13–16]. After generating necessary features from the protein sequences adopted from Bigram-PGK [11], a cost-sensitive SVM [14, 17–19] classifier has been used to predict phosphoglycerylation sites by minimizing class-level imbalance in benchmark dataset. The workflow of our proposed predictor is shown in Fig 1. For validating the statistical significance of the results, 10-fold cross-validation has been repeated ten times, and the average performances of each evaluation metric have been reported in the Results section. It can be observed that our proposed predictor, predPhogly-Site has achieved superior prediction performance than all the existing predictors. The attained performance of predPhogly-Site in terms of specificity, sensitivity, precision, accuracy, MCC, and AUC are 99.97%, 100%, 99.20%, 99.97%, 99.58%, and 99.99%, respectively. The promising results obtained by predPhogly-Site indicates that it can be used as a high-throughput supporting tool for phosphoglycerylation site prediction.

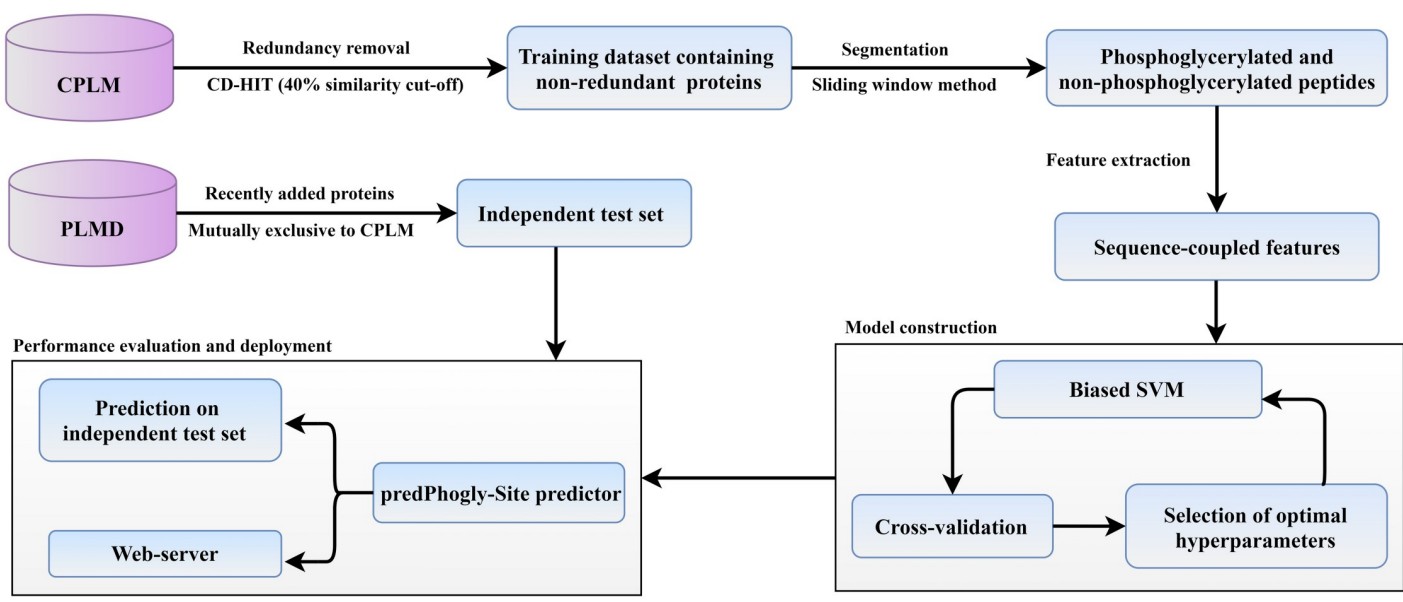

**Fig 1. An overview of predPhogly-Site for phosphoglycerylation site prediction.**

Highlighted in a series of recently published predictors [3, 6, 14, 19–23], to develop an efficient predictor with regards to computational biology, one should go through Chou's five-step [14, 24, 25] guidelines: i) generating an acceptable benchmark dataset for training and testing the system, ii) formulating the sequences using proper mathematical representations, iii) developing a prediction approach or introducing a robust prediction algorithm, iv) conducting rigorous cross-validation tests to evaluate predictive accuracy, and v) providing an accessible and easy-to-use web-server. Following these steps, details of materials, methods, results, and analysis will be discussed in the following sections.

## Materials and methods

### Dataset

In this study, verified annotations of phosphoglycerylation sites were obtained from the CPLM version 2.0 [26], one of the reliable repositories of post-translational modification in lysine residue, and corresponding protein sequences were retrieved from UniProt knowledge-base [27] for developing the prediction model. Subsequently, redundant sequences were discarded with 40% similarity cutoff using CD-HIT [28] for avoiding bias in performance evaluation as this level of redundancy removal was widely accepted [11, 24, 29, 30]. As a result, a total of 91 non-redundant proteins were held out for constructing a benchmark dataset. There were 111 experimentally annotated phosphoglycerylated sites and 3249 non-phosphoglycerylated sites, which was identical to the most recent predictor, Bigram-PGK's [11] dataset (see Table 1). The benchmark dataset containing protein sequences and site positions are given in S1 File. An overview of the dataset preparation as part of the prediction model development is presented

**Table 1. Summary of the non-redundant phosphoglycerylation dataset.**

| Similarity threshold | No. of non-redundant proteins | Phosphoglycerylated sites | Non-phosphoglycerylated sites |
|---|---|---|---|
| 40% | 91 | 111 | 3249 |

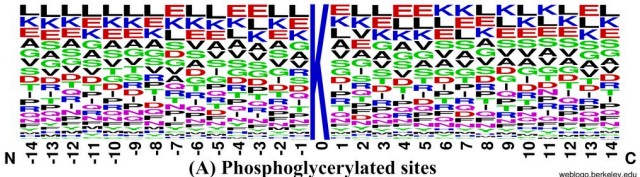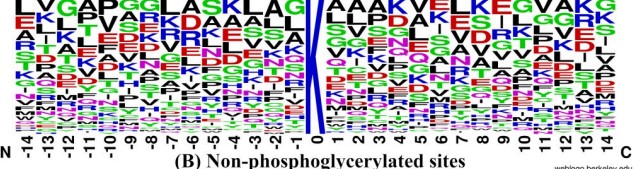

**Fig 2. Amino acid frequencies around the K-PTM and non-K-PTM sites.**

in Fig 1. For verifying the statistically significant difference among the positive and negative sites in the obtained dataset, the distribution of amino acid residues in the phosphoglycerylated sites and non-phosphoglycerylated sites are visually analyzed with the help of WebLogo [31] (see Fig 2A and 2B).

To demonstrate the viability of the proposed predictor predPhogly-Site for new proteins, an independent test set was constructed with recent phosphoglycerylation sites, utterly unknown to the benchmark dataset used for prediction model development. Protein sequences with recent phosphoglycerylation sites were collected from the PLMD database [32] (version 3.0), which is an upgraded version of the CPLM database [26], released nearly 03 years later with many newly discovered PTM sites. For ensuring the non-existence of training proteins in the independent test set, we considered only those proteins which were newly added to the PLMD repository much after the creation of the benchmark dataset with verified phosphoglycerylation sites. Therefore, we obtained 33 proteins with 41 phosphoglycerylated sites and 1334 non-phosphoglycerylated sites for the independent test (available as S2 File). Furthermore, the non-existence of recent test sites was verified manually for avoiding accidental bias in performance benchmarking.

### Feature construction

To formulate the phosphoglycerylation site sequences more meticulously and comprehensively, Chou's scheme [9, 13, 33] was adopted. According to this scheme, a potential phosphoglycerylation site containing sequence fragment could be expressed as:

$$\Theta_\zeta(K) = Q_1 Q_2 \ldots Q_{\zeta-1} Q_\zeta K Q_{\zeta+1} Q_{\zeta+2} \ldots Q_{2\zeta-1} Q_{2\zeta} \tag{1}$$

Where $Q_1$ to $Q_\zeta$ denote the leftward and $Q_{\zeta+1}$ to $Q_{2\zeta+1}$ denote the rightward amino acid residues, respectively, while $\zeta$ being an integer and centered 'K' indicating "lysine" [14]. Furthermore, the peptide sequences $\Theta_\zeta(K)$ can be categorized into two types: $\Theta_\zeta^+(K)$ and $\Theta_\zeta^-(K)$, where the first one denotes phosphoglycerylated peptide and the later one denotes non-phosphoglycerylated peptide with a lysine residue at its center [9, 14]. The sliding window method [9] was adopted to segment the phosphoglycerylation protein sequences with different window size where $\zeta = 1, 2, 3, \ldots 32$. Based on the MCC value, window size was selected as $(2\zeta + 1) = 29$ where $\zeta = 14$ (i.e. 14 rightstream and 14 leftstream amino acid residues). It should be mentioned that, only the window sizes less than 65 were taken under consideration due to the compelling protein sequence length [11]. With a sequence fragment of window size 29, Eq (1) could be expressed as:

$$\Theta(K) = Q_1 Q_2 \ldots Q_{13} Q_{14} K Q_{15} Q_{16} \ldots Q_{27} Q_{28} \tag{2}$$

At the time of segmentation, for making site sequences' of equal length, the lacking amino acids were filled with 'X' residue [9, 34]. As a result, the phosphoglycerylation dataset had

taken the following form:

$$S_\zeta(K) = S_\zeta^+(K) \cup S_\zeta^-(K) \tag{3}$$

where the positive subset $S_\zeta^+(K)$ could contain only $\Theta_\zeta^+(K)$ samples, while the negative subset $S_\zeta^-(K)$ could contain only $\Theta_\zeta^-(K)$ samples with their center residue $K$. All the segmented sequences with the expression of Eqs (2) and (3) are provided in S1 File.

For extracting pertinent features hidden in amino acid sequences, different sequence encoding methods such as amino acid composition, pseudo amino acid composition were used initially. However, in the proposed predictor predPhogly-Site, the vectorized sequence-coupled model [3, 14–16, 35] has been incorporated into general PseAAC [3, 14, 33, 35–39] to extract features from the phosphoglycerylation sites conserving the sequence pattern information. According to this conception, the peptide sample in Eq (2) can be expressed as:

$$\Theta(K) = \Theta^+(K) - \Theta^-(K) \tag{4}$$

where,

$$\Theta^+(K) = \begin{bmatrix} \Theta^+(Q_1|Q_2) \\ \Theta^+(Q_2|Q_3) \\ \vdots \\ \Theta^+(Q_{13}|Q_{14}) \\ \Theta^+(Q_{14}) \\ \Theta^+(Q_{15}) \\ \Theta^+(Q_{16}|Q_{15}) \\ \vdots \\ \Theta^+(Q_{27}|Q_{26}) \\ \Theta^+(Q_{28}|Q_{27}) \end{bmatrix} \qquad \Theta^-(K) = \begin{bmatrix} \Theta^-(Q_1|Q_2) \\ \Theta^-(Q_2|Q_3) \\ \vdots \\ \Theta^-(Q_{13}|Q_{14}) \\ \Theta^-(Q_{14}) \\ \Theta^-(Q_{15}) \\ \Theta^-(Q_{16}|Q_{15}) \\ \vdots \\ \Theta^-(Q_{27}|Q_{26}) \\ \Theta^-(Q_{28}|Q_{27}) \end{bmatrix} \tag{5}$$

where, $\Theta^+(Q_1|Q_2)$ denotes the conditional probability of amino acid $Q_1$ at the leftmost position given that its adjacent right member is $Q_2$ and the same applies for remaining indices of leftward residues [24]. Similarly, $\Theta^+(Q_{28}|Q_{27})$ denotes the conditional probability of amino acid $Q_{28}$ at the rightmost position given that its adjacent left member is $Q_{27}$ and so forth. In contrast, only $\Theta^+(Q_{14})$ and $\Theta^+(Q_{15})$ are of non-conditional probability as K is the adjoining member of both amino acids $Q_{14}$ and $Q_{15}$ [3, 6, 14, 15, 24]. In order to calculate the probability values of $\Theta^+(Q_{14})$ and $\Theta^+(Q_{15})$, firstly, we have to find the frequency of a given amino acid $Q_{14}$ and $Q_{15}$ from the set of phosphoglycerylated peptides [15]. Then the obtained values should be divided by the frequency of all amino acids occurring at position 14 and 15 respectively. Accordingly, $\Theta^-(K)$ in Eq (5), with its probabilistic components could also be deduced from the set of non-phosphoglycerylated peptides. A few literature on vectorized sequence-coupling model [3, 13, 15, 16] could provide a better understanding of the procedure of probability calculation out of any dataset. Finally, a 28-dimensional feature vector was obtained by using Eqs 4 and 5 for each potential phosphoglycerylated and non-phosphoglycerylated sample.

For better visualization and insights on the sequence-coupling effects at different positions of any sample, we have stored all possible combinations of conditional probability values extracted from the positive subset i.e. $\Theta^+(Q_1|Q_2)$ to $\Theta^+(Q_{13}|Q_{14})$ and $\Theta^+(Q_{16}|Q_{15})$ to $\Theta^+(Q_{28}|Q_{27})$ in one data frame (available in S3 File) and non-conditional probability values for each

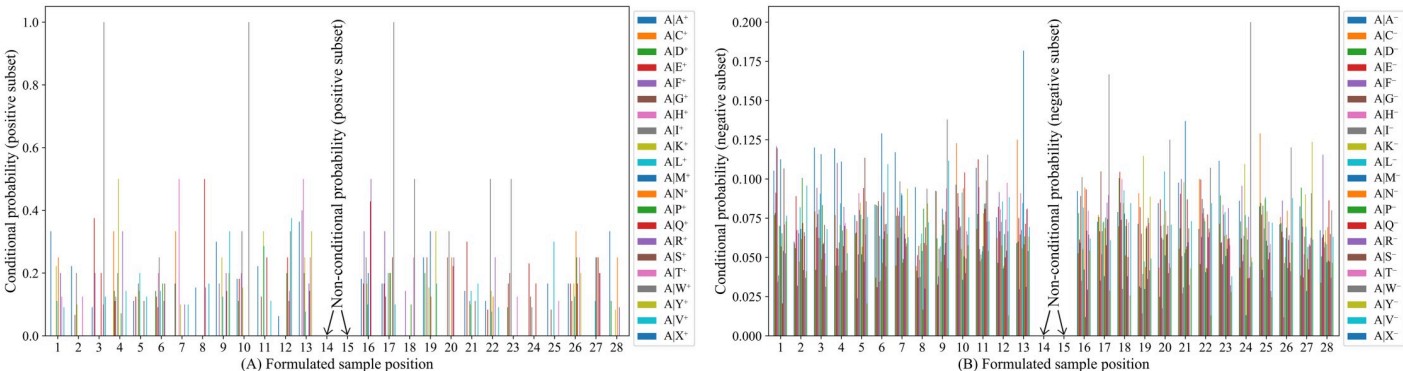

**Fig 3. The conditional probability of amino acids at sample positions 1 to 13 and 15 to 28.**

amino acid residue extracted from the positive subset i.e. $\Theta^+(Q_{14})$ and $\Theta^+(Q_{15})$ in another data frame (available in S4 File) using Pandas library [40], where the columns represent the formulated sample positions and the rows represent the amino acid residues. It should be mentioned that there could be $21 \times 21 = 441$ (including the dummy amino acid residue $'X'$) possible combinations of conditional probability values and 21 non-conditional probability values [15] for each position at any formulated sample. Similarly, the conditional and non-conditional probability values extracted from the negative subset are stored in two separate data frames and provided in S3 and S4 Files, respectively. Fig 3A depicts the conditional probability values of amino acid residue $'A'$ which have been calculated from the positive subset, given that its right member is any of the 21 amino acid residues at sample positions 1 to 13 and the conditional probability values of any of the 21 amino acid residue given that the left member is $'A'$ at sample positions 16 to 28. Similarly, Fig 3B depicts the conditional probability values of amino acid residue $'A'$ which have been calculated from the negative subset, given that its right member is any of the 21 amino acid residues at sample positions 1 to 13 and the conditional probability values of any of the 21 amino acid residue given that the left member is $'A'$ at sample positions 16 to 28. The non-conditional probability values of 21 amino acid residues derived from the positive subset at sample positions 14 and 15 are illustrated in Fig 4A and The non-conditional probability values of 21 amino acid residues derived from the negative subset at sample position 14 and 15 are shown in Fig 4B.

## Prediction method and addressing data imbalance

Phosphoglycerylation site prediction problem defined in the previous section is a classification problem. Statistical learning algorithms such as k-nearest neighbor [41], random forest [42] which are widely used in different bioinformatic prediction model development, support vector machine (SVM) [43, 44] is one of the dominant and successful among these algorithms [24, 45]. Apart from that, the structural risk minimization involves a biasing problem where the majority class [24, 46] influences the classification weight. As the set of phosphoglycerylation peptides was highly skewed (i.e. the ratio between positive and negative peptides was approximately 1:29), it could affect the classification model training directly. Inspired by the success of biasing internal decision function during training, as highlighted in recent research [8, 14, 17, 19], different penalty costs $C^+$, and $C^-$ were assigned for phosphoglycerylated sites and non-phosphoglycerylated sites, respectively for addressing imbalance issue. Therefore, SVM with cost-sensitivity was applied as a core learning algorithm for prediction model

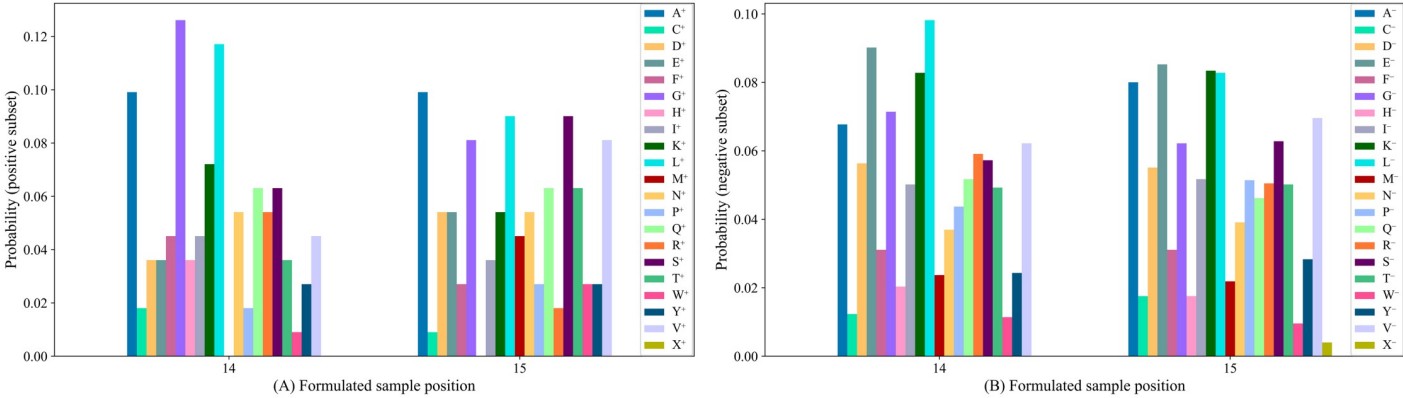

**Fig 4. Probabilistic information of 21 amino acids at sample positions 14 and 15.**

development which can be formulated as:

$$min_{w,\xi} \frac{1}{2}\|w\|^2 + C^+ \sum_{k=1}^{q} \xi_k + C^- \sum_{k=q+1}^{n} \xi_k \qquad (6)$$

(Subject to: $Y_k(w.\varphi(X_k) + a) \geq 1 - \xi_k$ for all, $k = 1, 2, .., n$)

where the training set is denoted by $\{(X_k, Y_k), k = 1, 2, \ldots, n\}$ and first q samples (i.e. $Y_k = 1$, $k = 1, 2, \ldots, q$) are assumed as the positive samples while the rest are assumed as the negative samples (i.e. $Y_k = -1$, $k = q + 1, q + 2, \ldots, n$). The non-linear feature mapping and slack variables are denoted by $\varphi(X)$ and $\xi_k(k = 1, 2, \ldots, n)$, respectively [45, 47]. In our experiments with SVM, as the kernel function, Gaussian RBF was adopted which can be described as: $\Upsilon(X_k, X_j) = \varphi(X_k)^T \varphi(X_j) = exp(-\gamma\|x_i - x_j\|^2)$, where $\gamma > 0$. However, for effective separation of positive and negative samples, addressing the class imbalance problem, misclassification costs $C^+ = \frac{C*n}{2*q}$ and $C^- = \frac{C*n}{2*(n-q)}$ were assigned for phosphoglycerylated sites and non-phosphoglycerylated sites, respectively.

## Formulation of evaluation metrics

To objectively assess the prediction performance of predPhogly-Site, we have utilized five widely used statistical metrics, such as accuracy (ACC), sensitivity (Sn), specificity (Sp), precision (pre) and Matthew's Correlation Coefficient (MCC) [20, 24, 30, 45, 47–52]. These matrices can be defined in terms of true positive (TP), false positive (FP), true negative (TN) and false negative (FN) prediction made by the predictor as following:

$$\begin{cases} Sn = \dfrac{TP}{TP + FN} \\ Sp = \dfrac{TN}{TN + FP} \\ Precision = \dfrac{TP}{TP + FP} \\ ACC = \dfrac{TP + TN}{TP + TN + FP + FN} \\ MCC = \dfrac{(TP \times TN) - (FP \times FN)}{\sqrt{(TP + FP)(TP + FN)(TN + FP)(TN + FN)}} \end{cases} \qquad (7)$$

To the best of our knowledge, state-of-the-art phosphoglycerylation site predictors [8, 9, 11, 12] have also estimated their performance based on these metrics. Thus, performance assessment using these metrics was essential to establish a fair comparative benchmarking. Eventually, we have considered the area under the ROC curve (AUC) [24, 53] in addition to MCC for illustrating the stability and robustness of the prediction model.

## Validation of the proposed model

To evaluate the statistical significance of a novel predictor's anticipated performance, three validation schemes, such as k-fold cross-validation, jackknife test, and independent test are widely used [14, 24]. Although the jackknife test can always draw out a unique result for a given dataset and highly desirable, to reduce the computational complexity of model development, researchers prefer k-fold cross-validation over the jackknife test for validating their PTM prediction models [8, 45]. Moreover, existing phosphoglycerylation site predictors validated their anticipated accuracy using k-fold cross validation except Phogly-PseAAC [9]. Even, the most recent predictor, Bigram-PGK [11] validated their model using 10-fold cross-validation and compared with existing predictors. Therefore, to develop and validate our proposed predictor predPhogly-Site, 10-fold cross-validation was adopted. However, as the 10-fold cross-validation involved some arbitrariness, highlighted in [9, 24], to validate the stability, it was repeatedly executed for 10 times. For finding the best performing predictor, a set of prediction models were generated for the hyperparameters $C$ and $\gamma$ within the grid of $C = \{2^0, 2^1, 2^2, \ldots, 2^8\}$ and $\gamma = \{2^{-1}, 2^{-2}, 2^{-3}, \ldots, 2^{-8}\}$. Using 10-fold cross-validation with 10 repeats, the best model with optimal hyperparameters $C$ and $\gamma$ were selected (see Table 2) depending on the demonstrated AUC.

The 10-iterations of 10-fold cross-validation were performed according to the following steps:

Step 1: Extract the sequence-coupled features from the segmented sequences provided in S1 File using Eqs (4) and (5).

Step 2: Divide the extracted dataset randomly into 10 disjoint sets.

Step 3: Select 1 set as test set and utilize the remaining 9 sets as training set.

Step 4: Train the RBF kernel based SVM predictor with the training set using the optimal hyperparameters $(C, \gamma)$ of the respective iteration (see Table 2).

Step 5: Perform prediction on the test set.

Step 6: Repeat steps 2 to 5 until all 10 sets had been used for testing.

Step 7: Merge the prediction outputs and measure the performance with Eq 7.

Step 8: Repeat steps 1 to 7 for 10 times.

**Table 2. Selected parameters of 10-fold cross validation (10 iterations).**

| Iteration | $1^{st}$ | $2^{nd}$ | $3^{rd}$ | $4^{th}$ | $5^{th}$ |
|---|---|---|---|---|---|
| $C$ | $2^0$ | $2^0$ | $2^0$ | $2^0$ | $2^0$ |
| $\gamma$ | $2^{-1}$ | $2^{-2}$ | $2^{-2}$ | $2^{-2}$ | $2^{-2}$ |
| Iteration | $6^{th}$ | $7^{th}$ | $8^{th}$ | $9^{th}$ | $10^{th}$ |
| $C$ | $2^1$ | $2^2$ | $2^2$ | $2^0$ | $2^0$ |
| $\gamma$ | $2^{-1}$ | $2^{-2}$ | $2^{-2}$ | $2^{-2}$ | $2^{-2}$ |

Step 9: Measure the average performance of 10 repetitions with corresponding standard deviations.

The predictive decision-making workflow of predPhogly-Site is available at https://github.com/Sabit-Ahmed/predPhogly-Site as a git repository. For additional validation, an independent test was performed on a set of recent phosphoglycerylation sites. It will be discussed thoroughly in the next section.

## Results and discussions

### Performance of predPhogly-Site

In this work, we employed SVM with variable cost adjustments [14, 19, 24] for suppressing the imbalance between phosphoglycerylated and non-phosphoglycerylated sites. For separating samples by transforming to higher dimensional feature space, radial basis kernel function [14, 22, 24] was utilized. The average results of the considered statistical performance measures with their standard deviations in 10 repeats are presented in Table 3. As shown in Table 3, the proposed prediction model could predict phosphoglycerylation sites with 99.97% accuracy. In addition to that, its sensitivity, specificity, MCC and AUC measure crossed a benchmark of 99%. Moreover, standard deviations were almost negligible in the case of all the measures. However, for constructing the proposed predictor predPhogly-Site to be deployed as a web service, the benchmark dataset and the prediction model's hyper-parameters with the highest AUC in 10 repetitions (i.e. $C = 2^0$ and $\gamma = 2^{-2}$) were used. An overview of establishing predPhogly-Site is depicted in Fig 1.

### Comparative analysis of cross-validation performance

To evaluate the effectiveness of the proposed predictor, predPhogly-Site, we compared it with four state-of-the-art phosphoglycerylation site predictors, such as Phogly-PseAAC [9], CKSAAP_PhoglySite [8], iPGK-PseAAC [12] and Bigram-PGK [11]. Among these predictors, the first three i.e. Phogly-PseAAC, CKSAAP_PhoglySite, and iPGK-PseAAC were benchmarked on the same phosphoglycerylation site dataset which was prepared by Xu et al. [9]. Prediction from Phogly-PseAAC and iPGK-PseAAC could be accessed by their web interface. Though CKSAAP_PhoglySite was also accessible by its Matlab interface, there was no such accessibility option in the most recent predictor, Bigram-PGK. However, Bigram-PGK had collected prediction results from these accessible predictors for its benchmark dataset and reported comparative outcomes for all the considered performance metrics. Thus, for conducting a fair comparison with all these predictors, our primary benchmark dataset, which was not resampled as Bigram-PGK's one, was submitted to the webserver of Phogly-PseAAC and iPGK-PseAAC for getting prediction outcomes. However, CKSAAP_PhoglySite's predictions were obtained through its Matlab interface. After achieving the prediction outcomes from the Phogly-PseAAC, CKSAAP_PhoglySite, and iPGK-PseAAC on the benchmark dataset constructed for this study, the corresponding performance was measured on the same validation set utilized for evaluating our predictor predPhogly-Site (see Section "Validation of the proposed model"). As we adopted different technique for handling the data imbalance issue

**Table 3. Cross-validation performance of predPhogly-Site on the benchmark dataset.**

| Predictor | Sp | Sn | Pre | ACC | MCC | AUC |
|---|---|---|---|---|---|---|
| predPhogly-Site | 0.9997 ± 0.0001 | 1.00±0.00 | 0.9920±0.0027 | 0.9997±0.0001 | 0.9958±0.0014 | 0.9999±0.00 |

**Table 4. Cross-validation performance of the existing prediction systems.**

| Predictor | Sp | Sn | Pre | ACC | MCC | AUC |
|---|---|---|---|---|---|---|
| iPGK-PseAAC | 0.9846 | 0.4595 | 0.5050 | 0.9673 | 0.4648 | 0.7220 |
| iPGK-PseAAC* | 0.9864 | 0.4555 | 0.9548 | 0.8119 | 0.5692 | 0.7230 |
| CKSAAP_PhoglySite | 0.8941 | 0.8288 | 0.2110 | 0.8920 | 0.3845 | 0.8615 |
| CKSAAP_PhoglySite* | 0.9420 | 0.8285 | 0.8765 | 0.9043 | 0.7818 | 0.8854 |
| Phogly-PseAAC | 0.7064 | 0.6937 | 0.0747 | 0.7060 | 0.1550 | 0.7000 |
| Phogly-PseAAC* | 0.7193 | 0.6927 | 0.5518 | 0.7102 | 0.3951 | 0.7062 |
| Bigram-PGK* | 0.8973 | 0.9642 | 0.8253 | 0.9193 | 0.8330 | 0.9306 |
| **predPhogly-Site** | **0.9997** | **1.00** | **0.9920** | **0.9997** | **0.9958** | **0.9999** |

* Corresponds to the experimental findings reported by the Bigram-PGK study [11].

and could not obtain the prediction outcomes from the Bigram-PGK predictor on our benchmark dataset, a comparative summary of all the measures was presented in Table 4 in line with Bigram-PGK's experimental findings [11]. As shown in Table 4 and Fig 5, predPhogly-Site achieved a significant improvement over Phogly-PseAAC, CKSAAP_PhoglySite, and iPGK-PseAAC on the same benchmark dataset used in this study. It remarkably outperformed these predictors in sensitivity, specificity, overall accuracy, and AUC. For instance, predPhogly-Site crossed the milestone of 99% in case of sensitivity, specificity, precision, overall accuracy, MCC and AUC.

However, the most recent predictor, Bigram-PGK's [11] performance was relatively higher in most of the metrics. It obtained a sensitivity of 96.42%, an accuracy of 91.93%, an MCC of 83.30%, and an AUC of 93.06% on the dataset utilized in Bigram-PGK [11]. As demonstrated in Table 4, our proposed predictor predPhogly-Site also outperformed Bigram-PGK [11] by 3.58% in sensitivity, 8.04% in accuracy measure, 16.28% in MCC and 6.93% in AUC.

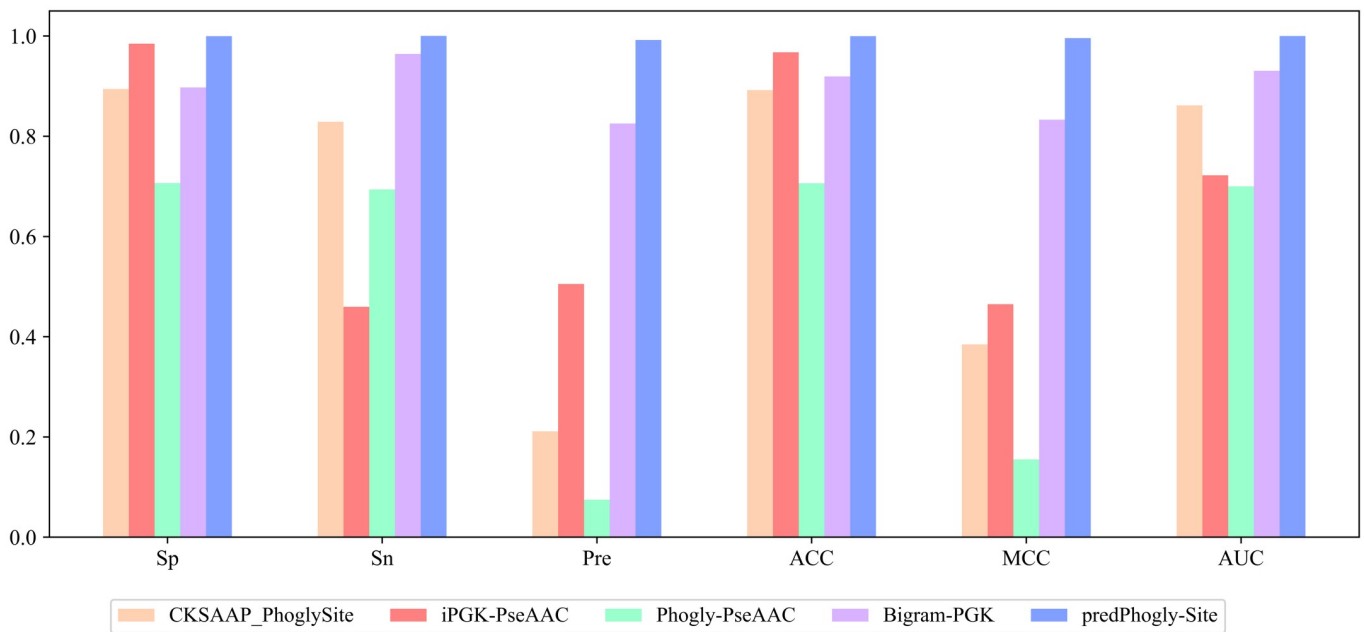

**Fig 5. Cross-validation performance of the available predictors.**

Furthermore, the effectiveness of predPhogly-Site over the recent predictors including Bigram-PGK [11] has been demonstrated in Fig 5.

It can be observed that a comparatively higher specificity and precision of 98.64% and 95.48%, respectively, were obtained by iPGK-PseAAC [12] on the Bigram-PGK's [11] resampled dataset. Our proposed predictor, predPhogly-Site, has obtained 1.33% and 3.72% increased performance in both specificity and precision, respectively. Both the results represented in Table 4 and Fig 5 indicate that our proposed predictor predPhogly-Site can identify phosphoglycerylation sites more effectively than any other existing predictors.

It is worth mentioning that among these predictors, Phogly-PseAAC [9] has employed the position-specific amino acid propensity which reflects the position-wise occurrence frequency of each amino acid and the K-Nearest Neighbor (KNN) algorithm for prediction, CKSAAP_PhoglySite [8] has utilized the composition of k-spaced amino acid pairs with the fuzzy SVM, iPGK-PseAAC [12] has applied the pairwise coupling technique with the posterior probability-based SVM and Bigram-PGK [11] have considered the SVM engine with the combination of position-specific scoring matrix and profile bigrams for performance improvement.

It might be intuitive to find some insight into why our proposed predictor predPhogly-Site achieved such superior performance. It was possible because of the effective representation of phosphoglycerylation modification in terms of sequence coupling model among the amino acid residues via the conditional probability (see Figs 3 and 4). Suppressing the imbalance ratio of phosphoglycerylated and non-phosphoglycerylated sites using different error costs based SVM also boosted up the performance improvement.

However, the precision calculation measures the believability of a system when it says a peptide sample is phosphoglycerylated. According to Eq 7, the precision measure depends highly on the false positive rate, and a lower false positive rate results in a higher precision rate. In the Bigram-PGK [11] study, the dataset contained only 111 positive samples and 224 negative samples after applying the k-nearest neighbor cleaning treatment [11] and the experimental findings on the resampled dataset might not reflect the false positive rate properly. Moreover, the existing predictors i.e. iPGK-PseAAC, CKSAAPPhoglySite, and Phogly-PseAAC might not handle the real world imbalanced situation of the dataset appropriately. Hence, when we have uploaded the benchmark dataset containing 111 positive instances and 3249 negative instances (see Table 1) to the web or Matlab interfaces of the existing predictors, the false positive rates have come out higher and results in lower precision rates as compared to the experimental findings reported by the Bigram-PGK study (see Table 4). On the other hand, our proposed predictor has obtained a much lower false positive rate and got a higher precision rate as well as higher sensitivity and specificity for having cost-sensitive SVM as an imbalance management technique. By observing all the performance measurements in this study, it can be concluded that our predictor predPhogly-Site could be a high throughput tool for predicting phosphoglycerylation sites more precisely.

### Independent test

Existing phosphoglycerylation site, particularly, the most recent predictor assessed their model using 10-fold cross-validation. However, some researchers [54–57] highlighted the necessity of independent test for assessing prediction model in addition to k-fold (e.g. k = 5,10) cross-validation. Thus, in our work, an independent test was conducted for further evaluation of our proposed model predPhogly-Site on an independent set of phosphoglycerylation sites. The same independent test set was uploaded to the web servers of the existing predictors i.e. iPGK-PseAAC, Phogly-PseAAC and predPhogly-Site for obtaining the prediction results.

**Table 5. Prediction performance in Independent test.**

| Predictor | Sp | Sn | Pre | ACC | MCC | AUC |
|---|---|---|---|---|---|---|
| iPGK-PseAAC | 0.9738 | 0.2927 | 0.2553 | 0.9535 | 0.2494 | 0.6332 |
| Phogly-PseAAC | 0.6837 | 0.6829 | 0.0622 | 0.6836 | 0.1329 | 0.6833 |
| CKSAAP_PhoglySite | 0.8823 | 0.7561 | 0.1649 | 0.8785 | 0.3161 | 0.8192 |
| **predPhogly-Site** | **0.9993** | **0.9512** | **0.9750** | **0.9978** | **0.9619** | **0.9752** |

However, the prediction results of CKSAAP_PhoglySite on the independent test set were obtained from the Matlab interface. The predictive performance of predPhogly-Site as well as other predictors were summarized in Table 5. However, as Bigram-PGK [11] had no established web-server, so we could not report the performance of these predictors on the independent test set.

As shown in Table 5, predPhogly-Site predicted independent phosphoglycerylation sites with specificity, sensitivity, precision, accuracy, MCC and AUC of 99.93%, 95.12%, 97.50%, 99.78%, 96.19% and 97.52%, respectively, which were almost identical to the cross-validation performance delineated in Table 4. According to the experimental results in Table 5 and the ROC curve illustrated in Fig 6, it was apparent that the proposed predictor predPhogly-Site achieved a significant improvement over their counterparts in terms of all the evaluation metrics.

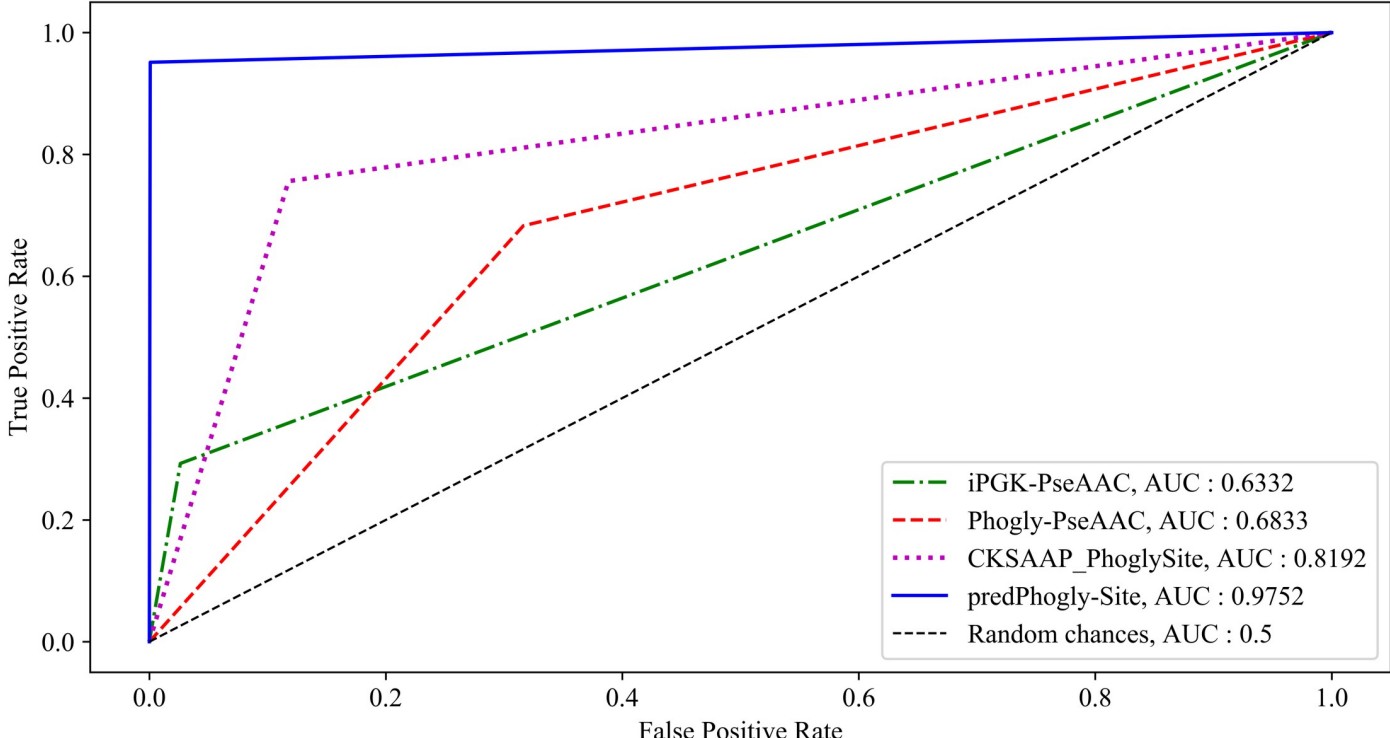

**Fig 6. Comparative ROC curves between different prediction methods based on the independent test.**

## Web-server

For intensifying user accessibility without the concern of experimental implementations, an easy-to-use web-server for predPhogly-Site has been developed. It can be accessed at http://103.99.176.239/predPhogly-Site. Users can submit one or more query protein sequence(s) directly on the web-server as text input in Fasta format or may prefer to upload as a batch to get their predictions. More detailed guidelines on how to use the web-server as well as the working mechanism of this server can also be found there. After submitting a query protein or as a batch, it may take a few moments to get the prediction result, depending on the availability of server resources. Finally, predPhogly-Site will generate a result page based on the user's submission, i.e., if protein sequences are submitted into the input box, the predictive data will be shown on the result page. Otherwise, it will be sent to the corresponding user through email.

## Conclusion

In this study, for identifying phosphoglycerylation sites in protein with higher accuracy, a novel computational tool, predPhogly-Site, has been developed utilizing the coupling effects in a sequence. It exploits probabilistic sequence pattern information with variable cost adjustment in the classifier's decision function for achieving higher predictive performance compared to the existing phosphoglycerylation site predictors. It has achieved significant performance improvement not only in the 10-fold cross-validation, which has been used as the benchmarking technique in the existing predictors but also in an independent test. Moreover, it has also achieved almost identical performance in both 10-fold cross-validation and independent test, which clearly demonstrates its stability. In the 10-fold cross-validation test, it has achieved more than 0.99 in both AUC and MCC, and in case of the independent test, it has achieved nearly 0.97 in the corresponding measures. These experimental outcomes demonstrate that predPhogly-Site is highly promising compared to the existing state-of-the-art phosphoglycerylation site predictors. It is expected to become a high throughput computational tool for PTM researcher for fast exploration of lysine modifications. Even the experimental scientists would be benefited from this web-based tool without going through its mathematical and implementation details. For further performance improvement and usability of this prediction tool, multiple types of post-translational modification with heterogeneous data would be incorporated simultaneously along with prediction interpretation support.

## Supporting information

**S1 File. Benchmark dataset.** The phosphoglycerylated proteins as well as the segmented sequences with respective protein ID and positions have been provided.
(PDF)

**S2 File. Independent test dataset.** Proteins which have been recently added to the PLMD database and completely unknown to the proposed system.
(PDF)

**S3 File. All possible combinations of the conditional probability values derived from the positive and negative subset.**
(XLSX)

**S4 File. The non-conditional probability values of 21 amino acids derived from the positive and negative subset.**
(XLSX)

## Author Contributions

**Conceptualization:** Sabit Ahmed, Afrida Rahman.

**Data curation:** Sabit Ahmed.

**Formal analysis:** Sabit Ahmed, Md Khaled Ben Islam, Julia Rahman.

**Investigation:** Afrida Rahman, Md. Al Mehedi Hasan, Md Khaled Ben Islam, Julia Rahman.

**Methodology:** Sabit Ahmed, Afrida Rahman.

**Resources:** Md. Al Mehedi Hasan, Shamim Ahmad.

**Software:** Afrida Rahman, Shamim Ahmad.

**Supervision:** Md. Al Mehedi Hasan, Shamim Ahmad.

**Validation:** Md. Al Mehedi Hasan, Md Khaled Ben Islam, Julia Rahman, Shamim Ahmad.

**Visualization:** Sabit Ahmed.

**Writing – original draft:** Sabit Ahmed, Afrida Rahman.

**Writing – review & editing:** Md Khaled Ben Islam, Julia Rahman.

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
