## [Decision Letter · Decision Letter 0]

14 Dec 2020

PONE-D-20-30897

predPhogly-Site: Predicting phosphoglycerylation sites by incorporating probabilistic sequence-coupling information into PseAAC and addressing data imbalance

PLOS ONE

Dear Dr. Ahmed,

Thank you for submitting your manuscript to PLOS ONE. After careful consideration, we feel that it has merit but does not fully meet PLOS ONE’s publication criteria as it currently stands. Therefore, we invite you to submit a revised version of the manuscript that addresses the points raised during the review process.

Both reviewers have major concerns about the paper. These include the preparation of the training and test sets and the methods. All points needs to be clarified. The writing and organisation of the paper also need clarification. 

We look forward to receiving your revised manuscript.

Kind regards,

Ozlem Keskin

Academic Editor

PLOS ONE

Journal Requirements:

Reviewers' comments:

Reviewer's Responses to Questions

**Comments to the Author**

1. Is the manuscript technically sound, and do the data support the conclusions?

Reviewer #1: Yes

Reviewer #2: Yes

2. Has the statistical analysis been performed appropriately and rigorously? 

Reviewer #1: Yes

Reviewer #2: Yes

3. Have the authors made all data underlying the findings in their manuscript fully available?

Reviewer #1: Yes

Reviewer #2: Yes

4. Is the manuscript presented in an intelligible fashion and written in standard English?

Reviewer #1: Yes

Reviewer #2: No

5. Review Comments to the Author

Reviewer #1: 1. The Authors have provided the training set in S1 File. It is not clear how a training set be fixed if 10-fold cross-validation is carried out where there was 10 disjoint test sets and then it was run for a total of 10 iterations. The train and test sets will be randomized each time cross-validation is carried out.

2. The independent test set is said to have 33 proteins. Are these proteins part of the 91 protein sequences obtained after applying the CD-HIT tool? How did the authors obtain the protein sequences for independent test set?

3. The Eq (4) is not very clear and why is there a minus sign between the two terms?

4. How exactly were the conditional as well as the non-conditional probabilities obtained?

5. The work has mentioned under the section ‘Comparative analysis of cross-validation performance’ that the predictions were obtained by submitting the benchmark dataset to the webservers, and also to the Matlab interface for one of the predictors. This however is not apparent as the same result from what Bigram-PGK obtained for those methods/predictors have been tabulated in Table 4. For fair comparison, the cross validation for other methods should also be undertaken for 10 iterations and averaged, while maintaining the same train and test sets in the respective iterations as with the predPhogly-Site method.

6. How did the Authors obtain their result on the independent test set? Was it obtained from the predPhogly-Site webserver? If not, how exactly was it obtained?

7. Why was the CKSAAP_PhoglySite left out in the Independent test? It was mentioned before that predictions could be obtained using its Matlab interface.

8. Calculation of AUC might not be possible if the probability scores are not present. If so, how did the Authors obtain AUC for iPGK-PseAAC and Phogly-PseAAC in Table 5?

9. If the predPhogly-Site is performing better in all the metrics, why is ‘almost all the metrics’ mentioned?

10. In conclusion, the Authors have mentioned that predPhogly-Site has been developed using ‘only primary sequence information’. This begs the question on how the conditional and non-conditional probabilities were obtained. If those probabilities were obtained using a tool such as PSI-BLAST toolbox (position specific scoring matrix of probabilities), it would be incorrect to use that sentence.

11. The Authors are requested to provide algorithm with train and test sets so that the method can be replicated in-order to verify the results.

12. Various PTMs are mentioned in the introduction, it would also be useful to mention about Glycation [PMID: 30717650].

Reviewer #2: Authors describe a computational tool predPhogly-Site for predicting phosphoglycerylation sites from protein sequences. predPhogly-Site extracts features from the protein sequences by a method adopted from the recent Bigram-PGK method and prediction is performed using support vector machines (SVM). The SVM is modified to handle class imbalance between positive dataset and negative dataset which inherently exist in this problem. The training dataset is taken from Bigram-PGK method in which the bias is avoided. The performance of predPhogly-Site tool is compared with those of the existing methods and predPhogly-Site tool comes out to outperform all of the existing methods. predPhogly-Site tool is also tested with an independent test set and again, its performance was the best. The tool is publicly available.

The method described in this paper has already been established and published. The data used in this study is same with Bigram-PGK. I presume that the only addition is the use of modified SVM to handle imbalanced data and this technique was previously formulated and published. The independent test set is also added. Finally, the performance seems to be better than those of existing methods and a web server is available for public access.

The structure of the manuscript should be improved.

Tables and Figures should be improved and better explained. For example, it is very hard to understand what Table 2 tells to the reader. Figures are given in low-resolution and the reader cannot understand them as they are. They should be better explained in the text and in their captions.

Sometimes the vocabulary is inconsistent. For example, “position-specific features” is used only once. What is “formulated samples”? In the beginning of “Feature Construction”, the authors use the terms “positive sites” and “negative sites”; then, these terms are not anymore used in that section.

In the lines 108-110, window size is written to be selected as 21 stating that it is based on preliminary analysis, however no reference is given for the preliminary analysis.

The authors may want to make a table of existing methods in which the columns may include dataset, method, performance values, etc for each method.

Spelling and typographical errors

Line 54

“1.00%” should be “100%”

The attained performance of predPhogly-Site in terms of accuracy, specificity, sensitivity, precision, MCC, and AUC are 99.86%, 99.86%, 1.00%, 95.94%, 54

97.88%, and 99.93%, respectively.

Lines 95-96

“are” is missing (but also it is difficult to understand the sentence)

“The non-existence of recent test sites also manually verified for avoiding accidental bias in performance benchmarking.”

6. PLOS authors have the option to publish the peer review history of their article (what does this mean?). If published, this will include your full peer review and any attached files.

Reviewer #1: No

Reviewer #2: No

---

## [Author Response · Author response to Decision Letter 0]

30 Dec 2020

Points raised by the academic editor:

and

Answer: Thank you very much for pointing out some issues regarding our manuscript. We have gone through the given links and changed our manuscript accordingly including the file naming format.

Points raised by the reviewers:

Reviewer #1: 

1. The Authors have provided the training set in S1 File. It is not clear how a training set be fixed if 10-fold cross-validation is carried out where there was 10 disjoint test sets and then it was run for a total of 10 iterations. The train and test sets will be randomized each time cross-validation is carried out.

Answer: Thank you very much for addressing the issue. We would like to inform you that the benchmark dataset provided in S1 File had been utilized for 10-fold cross-validation by following the steps mentioned in the section “Validation of the proposed model”. Furthermore, we had repeated it 10 times for achieving more stability in the performance measurement. As in the previous study (i.e. Bigram-PGK), the 10 fold cross-validation had been conducted for one time, we had reported the average performance of 10 iterations of 10-fold cross-validation in our study. Each time the train and test sets had been randomized as described in the same section. Later, after getting the best hyperparameters and statistical measures, the entire benchmark dataset was used to train the web-server.

2. The independent test set is said to have 33 proteins. Are these proteins part of the 91 protein sequences obtained after applying the CD-HIT tool? How did the authors obtain the protein sequences for independent test set?

Answer: Apparently, the 33 proteins were not part of the 91 protein sequences. Those were collected from the Protein Lysine Modifications Database (PLMD). As mentioned in Section “Dataset”, we considered the proteins which were newly added to this repository much after the creation of Bigram-PGK’s dataset. Furthermore, we had cross-checked those proteins manually and ensured that those have not existed in the benchmark dataset.

3. The Eq (4) is not very clear and why is there a minus sign between the two terms?

Answer: According to the sequence-coupling model mentioned in [1], [2], [3], Eq (4) was used to subtract the conditional and non-conditional probability values of the negative subset from the conditional and non-conditional probability values of the positive subset.

4. How exactly were the conditional as well as the non-conditional probabilities obtained?

Answer: We have provided a detailed discussion on how to calculate the conditional and non-conditional probability values out of any dataset in the section “Feature construction”. Additionally, we have provided references of a few established predictors where the vectorized sequence-coupled model [1], [2], [3], [4], [5] have been adopted. The notations of Eq (4) and Eq (5) have been changed in an easier to understand form in the revised manuscript.

5. The work has mentioned under the section ‘Comparative analysis of cross-validation performance’ that the predictions were obtained by submitting the benchmark dataset to the web-servers and also to the Matlab interface for one of the predictors. This however is not apparent as the same result from what Bigram-PGK obtained for those methods/predictors have been tabulated in Table 4. For a fair comparison, the cross-validation for other methods should also be undertaken for 10 iterations and averaged, while maintaining the same train and test sets in the respective iterations as with the predPhogly-Site method.

Answer: We have addressed this issue and reported the corresponding prediction results on both the dataset used in predPhogly-Site and Bigram-PGK in Section “Comparative analysis of cross-validation performance”. As we could not obtain the prediction outcomes from Bigram-PGK, we have compared the other three predictors’ (i.e. CKSAAP_PhoglySite, iPGK-PseAAC and Phogly-PseAAC) performance with the same benchmark dataset used in the predPhogly-Site study maintaining the same train and test folds on each iteration. Additionally, we have included the prediction performance of each predictor obtained by Bigram-PGK with different notations in Table 4. 

6. How did the Authors obtain their result on the independent test set? Was it obtained from the predPhogly-Site webserver? If not, how exactly was it obtained?

Answer: The independent test result was obtained from the predPhogly-Site web-server and has been mentioned in Section “Independent test”.

7. Why was the CKSAAP_PhoglySite left out in the Independent test? It was mentioned before that predictions could be obtained using its Matlab interface.

Answer: As mentioned in the supporting information of CKSAAP_PhoglySite, its execution needed a 32-bit Matlab package. We could not manage the 32-bit software at the time of our manuscript submission. Later, we have managed it and reported the prediction performance of CKSAAP_PhoglySite on the training set and independent test set in Table 4 and Table 5 respectively.

8. Calculation of AUC might not be possible if the probability scores are not present. If so, how did the Authors obtain AUC for iPGK-PseAAC and Phogly-PseAAC in Table 5?

Answer: We have followed a similar procedure utilized by Bigram-PGK for calculating AUC. This approach could be found at https://github.com/abelavit/Bigram-PGK, a GitHub link provided by Bigram-PGK.

9. If the predPhogly-Site is performing better in all the metrics, why is ‘almost all the metrics’ mentioned?

Answer: As we could not report the performance of CKSAAP_PhoglySite at that time, we could not state that our predictor performed the best among all the predictors. This issue has been resolved after obtaining the prediction performance from the CKSAAP_PhoglySite predictor and the text “almost all the metrics” has been corrected.

10. In conclusion, the Authors have mentioned that predPhogly-Site has been developed using ‘only primary sequence information’. This begs the question on how the conditional and non-conditional probabilities were obtained. If those probabilities were obtained using a tool such as PSI-BLAST toolbox (position specific scoring matrix of probabilities), it would be incorrect to use that sentence.

Answer: The probability values were calculated according to the vectorized sequence-coupling model proposed by K.C. Chou [1], [2], [3], [4], [5]. According to this formula, Eq (4) and Eq (5) were written. We have tried to simplify the equation and explained both in the manuscript in the Section “Feature construction” and in the response to question no. 3. The selection of the words in Section “Conclusion” might be inappropriate, and so we have mitigated this issue with proper sets of words in the revised manuscript.

11. The Authors are requested to provide algorithm with train and test sets so that the method can be replicated in-order to verify the results.

Answer: Thank you for your suggestion. Based on your suggestion, the steps of the cross-validation procedure have been provided in Section “Validation of the proposed model”.

12. Various PTMs are mentioned in the introduction, it would also be useful to mention about Glycation [PMID: 30717650].

Answer: Recognizing the importance of PTM site Glycation, we have mentioned it with proper reference. Thanks for providing valuable suggestions that could help to improve our revised manuscript.

References:

1. Chou, Kuo-Chen. "A vectorized sequence-coupling model for predicting HIV protease cleavage sites in proteins." Journal of Biological Chemistry 268.23 (1993): 16938-16948.

2. Chou, Kuo-Chen. "Prediction of human immunodeficiency virus protease cleavage sites in proteins." Analytical biochemistry 233.1 (1996): 1-14.

3. Qiu, Wang-Ren, et al. "iPTM-mLys: identifying multiple lysine PTM sites and their different types." Bioinformatics 32.20 (2016): 3116-3123.

4. Chou, Kuo‐Chen. "Prediction of protein cellular attributes using pseudo‐amino acid composition." Proteins: Structure, Function, and Bioinformatics 43.3 (2001): 246-255.

5. Chou, Kuo-Chen. "Some remarks on protein attribute prediction and pseudo amino acid composition." Journal of theoretical biology 273.1 (2011): 236-247.

Reviewer #2: Authors describe a computational tool predPhogly-Site for predicting phosphoglycerylation sites from protein sequences. predPhogly-Site extracts features from the protein sequences by a method adopted from the recent Bigram-PGK method and prediction is performed using support vector machines (SVM). The SVM is modified to handle class imbalance between positive dataset and negative dataset which inherently exist in this problem. The training dataset is taken from Bigram-PGK method in which the bias is avoided. The performance of predPhogly-Site tool is compared with those of the existing methods and predPhogly-Site tool comes out to outperform all of the existing methods. predPhogly-Site tool is also tested with an independent test set and again, its performance was the best. The tool is publicly available.

The method described in this paper has already been established and published. The data used in this study is same with Bigram-PGK. I presume that the only addition is the use of modified SVM to handle imbalanced data and this technique was previously formulated and published. The independent test set is also added. Finally, the performance seems to be better than those of existing methods and a web server is available for public access.

The structure of the manuscript should be improved.

Tables and Figures should be improved and better explained. For example, it is very hard to understand what Table 2 tells to the reader. Figures are given in low-resolution and the reader cannot understand them as they are. They should be better explained in the text and in their captions.

Sometimes the vocabulary is inconsistent. For example, “position-specific features” is used only once. What is “formulated samples”? In the beginning of “Feature Construction”, the authors use the terms “positive sites” and “negative sites”; then, these terms are not anymore used in that section.

In the lines 108-110, window size is written to be selected as 21 stating that it is based on preliminary analysis, however no reference is given for the preliminary analysis.

The authors may want to make a table of existing methods in which the columns may include dataset, method, performance values, etc for each method.

Spelling and typographical errors

Line 54

“1.00%” should be “100%”

The attained performance of predPhogly-Site in terms of accuracy, specificity, sensitivity, precision, MCC, and AUC are 99.86%, 99.86%, 1.00%, 95.94%, 54

97.88%, and 99.93%, respectively.

Lines 95-96

“are” is missing (but also it is difficult to understand the sentence)

“The non-existence of recent test sites also manually verified for avoiding accidental bias in performance benchmarking.”

Answer: Thank you very much for addressing some important issues and providing your valuable suggestions associated with them. We have tried to alleviate these issues in the revised version of our manuscript.

1. The structure of the manuscript should be improved.

Tables and Figures should be improved and better explained. For example, it is very hard to understand what Table 2 tells to the reader. Figures are given in low-resolution and the reader cannot understand them as they are. They should be better explained in the text and in their captions. 

Answer: We have tried our best to improve the overall structure of our manuscript. The Tables and Figures have been taken under consideration and we have tried to provide a proper explanation with captions. The column names of Table 2 have been modified so that it can better explain what purpose it serves. In the revised manuscript, we have tried to provide high-resolution Figures which is further corrected by PACE which helps to ensure that Figures meet PLOS requirements.

2. Sometimes the vocabulary is inconsistent. For example, “position-specific features” is used only once. What is “formulated samples”? In the beginning of “Feature Construction”, the authors use the terms “positive sites” and “negative sites”; then, these terms are not anymore used in that section.

Answer: We have revised the manuscript and tried our best to reduce the inconsistency of vocabulary throughout the manuscript. The addressed issues have been resolved by using proper sets of words, especially, at the beginning of Section “Feature construction”. In addition to that, we would like to inform you that, we have obtained a set of formulated samples after adopting Chou’s scheme for sample formulation.

3. In the lines 108-110, window size is written to be selected as 21 stating that it is based on preliminary analysis, however no reference is given for the preliminary analysis.

Answer: Thank you very much for addressing such an important point. We would like to inform you that, previously we had considered the sliding window method with a window size of 3,5,7,9,….,21 and at the window size of 21, we have achieved much higher performance. However, based on your concern, we have experimented further to find the optimal window size. We have found that window size 29 gives the most promising results. We have reported the improved performance and reflected all the changes based on your addressed issues.

4. The authors may want to make a table of existing methods in which the columns may include dataset, method, performance values, etc for each method.

Answer: Thank you for the suggestion. We have tried to follow your suggestion by providing the performance of each method on the dataset constructed in the predPhogly-Site study and on the dataset of Bigram-PGK in Table 4 with distinct notations.

5. Spelling and typographical errors

Line 54

“1.00%” should be “100%”

The attained performance of predPhogly-Site in terms of accuracy, specificity, sensitivity, precision, MCC, and AUC are 99.86%, 99.86%, 1.00%, 95.94%, 97.88%, and 99.93%, respectively.

Answer: We have corrected the spelling and typographical error in Line 54. In addition to that, we have tried to find out these types of errors throughout the manuscript and made proper corrections.

6. Lines 95-96 

“are” is missing (but also it is difficult to understand the sentence)

“The non-existence of recent test sites also manually verified for avoiding accidental bias in performance benchmarking.”

Answer: We have made corrections in Lines 95-96 and provided easier to understand explanation. We would like to thank you once again for reviewing our manuscript and providing the necessary suggestions.

---

## [Decision Letter · Decision Letter 1]

20 Jan 2021

PONE-D-20-30897R1

predPhogly-Site: Predicting phosphoglycerylation sites by incorporating probabilistic sequence-coupling information into PseAAC and addressing data imbalance

PLOS ONE

Dear Dr. Ahmed,

Thank you for submitting your manuscript to PLOS ONE. After careful consideration, we feel that it has merit but does not fully meet PLOS ONE’s publication criteria as it currently stands. Therefore, we invite you to submit a revised version of the manuscript that addresses the points raised during the review process.

We look forward to receiving your revised manuscript.

Kind regards,

Ozlem Keskin

Academic Editor

PLOS ONE

Reviewers' comments:

Reviewer's Responses to Questions

**Comments to the Author**

1. If the authors have adequately addressed your comments raised in a previous round of review and you feel that this manuscript is now acceptable for publication, you may indicate that here to bypass the “Comments to the Author” section, enter your conflict of interest statement in the “Confidential to Editor” section, and submit your "Accept" recommendation.

Reviewer #1: (No Response)

Reviewer #2: All comments have been addressed

2. Is the manuscript technically sound, and do the data support the conclusions?

Reviewer #1: Yes

Reviewer #2: Yes

3. Has the statistical analysis been performed appropriately and rigorously? 

Reviewer #1: Yes

Reviewer #2: Yes

4. Have the authors made all data underlying the findings in their manuscript fully available?

Reviewer #1: Yes

Reviewer #2: Yes

5. Is the manuscript presented in an intelligible fashion and written in standard English?

Reviewer #1: Yes

Reviewer #2: Yes

6. Review Comments to the Author

Reviewer #1: The authors have addressed majority of the comments. However, a few comments were not addressed properly.

The authors have not provided algorithm with train and test sets so that the method can be replicated in-order to verify the results. The proposed method has achieved very high results and so it would be crucial to check and verify their result through the algorithm they have used.

New Comment: The authors are requested to check the Precision calculation as it seems to be quite low for the other methods in Table 4 when compared to what Bigram-PGK has reported for the methods. The other metrics appear to be similar.

Also, the references [1]-[5] are unnecessary and do not add any information.

Reviewer #2: The authors have appropriately answered my comments. The manuscript structure and content have improved and I support publishing the work.

7. PLOS authors have the option to publish the peer review history of their article (what does this mean?). If published, this will include your full peer review and any attached files.

Reviewer #1: No

Reviewer #2: No

---

## [Author Response · Author response to Decision Letter 1]

8 Feb 2021

Points raised by the reviewers:

Reviewer #1: 

1. The authors have addressed majority of the comments. However, a few comments were not addressed properly.

The authors have not provided algorithm with train and test sets so that the method can be replicated in-order to verify the results. The proposed method has achieved very high results and so it would be crucial to check and verify their result through the algorithm they have used.

Answer: Thank you very much for your feedback. We have tried addressing the comments which might not be covered in our previous submission. We have updated the flowchart for more insights on the overall procedure of constructing the predPhogly-Site predictor. It is mentioned in the “Introduction” section as well as in the “Dataset” section. Summarizing the steps included in our system, firstly, we have constructed the benchmark dataset from the CPLM database. The phosphoglycerylated proteins, as well as the segmented sequences with respective protein ID and positions, have been provided as File S1. Secondly, we have extracted the sequence-coupling features from the segmented sequences given in File S1. Then we have used 10 times 10-fold cross-validation scheme to train and evaluate our SVM based predictor with the optimal hyperparameters. The step-by-step guidelines are discussed in the “Validation of the proposed model” section and the optimal hyperparameters are given in Table 2. Later, we have performed an independent test where the test dataset contains newly added proteins (available as File S2), completely unknown to the predPhogly-Site predictor for further evaluation. We hope that our system is now reproducible and our results can be checked and verified easily.

2. The authors are requested to check the Precision calculation as it seems to be quite low for the other methods in Table 4 when compared to what Bigram-PGK has reported for the methods. The other metrics appear to be similar.

Answer: Thanks for your suggestion regarding the precision calculation. We have rechecked the precision calculation and explained this issue properly in Section “Comparative analysis of cross-validation performance” which reflects that the dataset used in the Bigram-PGK study contained only 111 positive samples and 224 negative samples and thus, the false-positive rates of the existing predictors i.e. iPGK-PseAAC, CKSAAPPhoglySite, and Phogly-PseAAC were lower and precision measures were higher. The Bigram-PGK dataset has not reflected the real future test data. Normally, in the future test dataset, the number of positive and negative sites will not be balanced. Moreover, the existing predictors i.e. iPGK-PseAAC, CKSAAPPhoglySite, and Phogly-PseAAC might not handle the real world imbalanced situation of the dataset appropriately. As a result, high false-positive rates and low precision measures were obtained on our benchmark dataset that contains 111 positive instances and 3249 negative instances.

3. Also, the references [1]-[5] are unnecessary and do not add any information.

Answer: We have removed a few references between [1]-[5] according to your suggestion. However, a few references have been still included as these references could be vital for further researches on phosphoglycerylation sites.

Reviewer #2: The authors have appropriately answered my comments. The manuscript structure and content have improved and I support publishing the work.

Answer: Thank you very much for your feedback.

---

## [Decision Letter · Decision Letter 2]

3 Mar 2021

PONE-D-20-30897R2

predPhogly-Site: Predicting phosphoglycerylation sites by incorporating probabilistic sequence-coupling information into PseAAC and addressing data imbalance

PLOS ONE

Dear Dr. Ahmed,

Thank you for submitting your manuscript to PLOS ONE. After careful consideration, we feel that it has merit but does not fully meet PLOS ONE’s publication criteria as it currently stands. Therefore, we invite you to submit a revised version of the manuscript that addresses the points raised during the review process. Please, respond to the reviewer's request for data availability.

We look forward to receiving your revised manuscript.

Kind regards,

Ozlem Keskin

Academic Editor

PLOS ONE

Journal Requirements:

Reviewers' comments:

Reviewer's Responses to Questions

**Comments to the Author**

1. If the authors have adequately addressed your comments raised in a previous round of review and you feel that this manuscript is now acceptable for publication, you may indicate that here to bypass the “Comments to the Author” section, enter your conflict of interest statement in the “Confidential to Editor” section, and submit your "Accept" recommendation.

Reviewer #1: (No Response)

2. Is the manuscript technically sound, and do the data support the conclusions?

Reviewer #1: Partly

3. Has the statistical analysis been performed appropriately and rigorously? 

Reviewer #1: Yes

4. Have the authors made all data underlying the findings in their manuscript fully available?

Reviewer #1: No

5. Is the manuscript presented in an intelligible fashion and written in standard English?

Reviewer #1: Yes

6. Review Comments to the Author

Reviewer #1: Major Revision.

Previous comments to the authors are not adequately addressed and hence I am summarising it again here.

The authors have stated that the result can be replicated by going through various section of the paper such as ‘Introduction’, ‘Dataset’, ‘Validation of the proposed model’, etc. However, for the review process, the authors are requested to put their dataset and code in a repository from which the reviewers can easily verify the result without the need to go about replicating the result by duplicating their method from scratch.

7. PLOS authors have the option to publish the peer review history of their article (what does this mean?). If published, this will include your full peer review and any attached files.

Reviewer #1: No

---

## [Author Response · Author response to Decision Letter 2]

8 Mar 2021

Journal Requirements:

Answer: Thanks to the concern very much for the valuable suggestion. I would like to inform you that we have revised and corrected the reference list according to the journal requirements. It is to be included that two of the references had been removed in our previously submitted manuscript according to the suggestion of one of the reviewers. The reviewer indicated that the first five references were irrelevant and unimportant even though there were some vital references in that list. It was not clear why those references seemed so insignificant to the reviewer. However, we had removed only reference numbers three and four as per the request of the reviewer. I would like to include the full citations which had been removed previously as well as in the current revised manuscript.

3.Weissman JD, Raval A, Singer DS. Assay of an intrinsic acetyltransferase activity of the transcriptional coactivator CIITA. In: Methods in enzymology. vol. 370. Elsevier; 2003. p. 378–386.

4. Chou KC. Impacts of bioinformatics to medicinal chemistry. Medicinal chemistry. 2015;11(3):218–234.

Points raised by the reviewers:

Reviewer #1: Major Revision. 

1. Previous comments to the authors are not adequately addressed and hence I am summarising it again here.

The authors have stated that the result can be replicated by going through various section of the paper such as ‘Introduction’, ‘Dataset’, ‘Validation of the proposed model’, etc. However, for the review process, the authors are requested to put their dataset and code in a repository from which the reviewers can easily verify the result without the need to go about replicating the result by duplicating their method from scratch.

Answer: Thank you for your kind response. We would like to inform you that a git repository containing the source code of predPhogly-Site is available at https://github.com/Sabit-Ahmed/predPhogly-Site, which could aid the reviewers in verifying the performance obtained by the predPhogly-Site predictor. All the relevant information i.e. the benchmark dataset, independent test dataset, extracted features, source code are provided either as supporting information or git repository and the results can be reproduced and verified. A web-server is also available at http://103.99.176.239/predML-Site for further verification.

---

## [Decision Letter · Decision Letter 3]

18 Mar 2021

predPhogly-Site: Predicting phosphoglycerylation sites by incorporating probabilistic sequence-coupling information into PseAAC and addressing data imbalance

PONE-D-20-30897R3

Dear Dr. Ahmed,

We’re pleased to inform you that your manuscript has been judged scientifically suitable for publication and will be formally accepted for publication once it meets all outstanding technical requirements.

Kind regards,

Ozlem Keskin

Academic Editor

PLOS ONE

Additional Editor Comments (optional):

Reviewers' comments:

Reviewer's Responses to Questions

**Comments to the Author**

1. If the authors have adequately addressed your comments raised in a previous round of review and you feel that this manuscript is now acceptable for publication, you may indicate that here to bypass the “Comments to the Author” section, enter your conflict of interest statement in the “Confidential to Editor” section, and submit your "Accept" recommendation.

Reviewer #1: All comments have been addressed

2. Is the manuscript technically sound, and do the data support the conclusions?

Reviewer #1: Yes

3. Has the statistical analysis been performed appropriately and rigorously? 

Reviewer #1: Yes

4. Have the authors made all data underlying the findings in their manuscript fully available?

Reviewer #1: Yes

5. Is the manuscript presented in an intelligible fashion and written in standard English?

Reviewer #1: Yes

6. Review Comments to the Author

Reviewer #1: authors addressed my concerns therefore i recommend to accept the paper in its current form.

7. PLOS authors have the option to publish the peer review history of their article (what does this mean?). If published, this will include your full peer review and any attached files.

Reviewer #1: **Yes: **Alok Sharma

---

## [Editor Report · Acceptance letter]

22 Mar 2021

PONE-D-20-30897R3 

predPhogly-Site: Predicting phosphoglycerylation sites by incorporating probabilistic sequence-coupling information into PseAAC and addressing data imbalance 

Dear Dr. Ahmed:

I'm pleased to inform you that your manuscript has been deemed suitable for publication in PLOS ONE. Congratulations! Your manuscript is now with our production department. 

Kind regards, 

on behalf of

Dr. Ozlem Keskin 

Academic Editor

PLOS ONE